# Evaluation of the Impact of Population Management on the Genetic Parameters of Selected Spiral-Horned Antelopes

**DOI:** 10.3390/biology13020104

**Published:** 2024-02-07

**Authors:** Ema Cetkovská, Karolína Brandlová, Rob Ogden, Barbora Černá Bolfíková

**Affiliations:** 1Faculty of Tropical AgriSciences, Czech University of Life Sciences Prague, Kamycka 129, 16500 Prague, Czech Republic; cetkovska@ftz.czu.cz; 2Royal (Dick) School of Veterinary Studies and the Roslin Institute, University of Edinburgh, Midlothian EH25 9RG, UK; rob.ogden@ed.ac.uk

**Keywords:** ex situ management, zoo populations, genetic diversity, Tragelaphini

## Abstract

**Simple Summary:**

Biodiversity is being lost at a rapid pace, with habitats being reduced and fragmented. Wild populations are declining, and zoos have become a crucial component of species conservation. However, zoo populations tend to be limited in size and based on few founders, which elevates the risk of genetic diversity loss. As the management needed to prevent such a risk is commonly reliant on incomplete and inaccurate pedigree data, the inclusion of molecular methods is recommended. This study assessed the relationship between the level of genetic diversity and management strategies, conservation status, demography, and geographic origin in European zoo populations of six spiral horned-antelope taxa using nuclear and mitochondrial DNA markers. We identified an association between the values of genetic diversity and the intensity of management, as well as the diversity and contribution of wild populations to founders of the captive stocks. Some zoo populations apparently consist of animals from genetically distinct lineages, which could have elevated the captive genetic diversity parameters but needs to be carefully considered as it can disrupt adaptive potential. The study shows that with careful interpretation, genetic data are very useful in the management of captive populations.

**Abstract:**

The rapid loss of biodiversity and the associated reduction and fragmentation of habitats means that ex situ populations have become an important part of species conservation. These populations, which are often established from a small number of founders, require careful management to avoid the negative effects of genetic drift and inbreeding. Although the inclusion of molecular data is recommended, their availability for captive breeding management remains limited. The aim of this study was to evaluate the relationship between the levels of genetic diversity in six spiral-horned antelope taxa bred under human care and their respective management strategies, conservation status, demography, and geographic origin, using 10 nuclear DNA microsatellite loci and mitochondrial control region DNA sequences. Our findings include associations between genetic diversity and management intensity but also with the diversity and contribution of wild populations to captive founders, with some populations apparently composed of animals from divergent wild lineages elevating captive genetic diversity. When population sizes are large, the potential advantages of maximizing genetic diversity in widely outcrossed populations may need careful consideration with respect to the potential disruption of adaptive diversity. Genetic data serve as a robust tool for managing captive populations, yet their interpretation necessitates a comprehensive understanding of species biology and history.

## 1. Introduction

Biodiversity is decreasing at an alarming rate worldwide, resulting not only in the extinction of certain species but also in significant reductions in genetic diversity of the species that persist [1]. Genetic diversity is a crucial component of biodiversity, which facilitates population adaptations to altering conditions and mitigates the risk of extinction [2,3]. This factor is especially important when facing the rapid rate of climate change [4]. Despite its importance, genetic diversity has historically been neglected in conservation policies and plans [5,6,7,8,9,10,11], with the predominant focus put on socioeconomically important taxa rather than diversity in wild populations [6,9,11,12]. Furthermore, loss of genetic diversity is not always immediately apparent [12] despite often having an earlier onset and faster progression than reductions in species diversity [12,13]. However, under the Global Biodiversity Framework [14], significantly greater emphasis is now placed on the conservation of diversity within species, thus demanding specific focus on understanding and protecting biodiversity at the level of population.

Genetic diversity loss is especially pronounced in small and fragmented populations [15] due to the absence of gene flow, increased levels of inbreeding, and the impact of genetic drift [16]. The average decline in vertebrate population abundance between 1970 and 2018 is estimated at 69% [17], with a mean genetic diversity loss of 6% [18]. As a result of the rapid rate of biodiversity degradation in natural populations, in situ conservation alone may be insufficient to sustain species [19], and the role of ex situ institutions, such as zoos, becomes increasingly important and sometimes crucial [20,21], especially for the protection of endangered species [19,20]. However, zoo populations are often founded by a small number of individuals and tend to be modest in size, thereby creating challenges for conservation genetic management [22].

Population management in zoos has included an explicit focus on genetic diversity for over twenty years, with methods based on pedigree data proving efficient in mitigating the diversity loss [23,24,25,26,27,28]. However, challenges often arise when pedigree records are incomplete or inaccurate [29,30]. Moreover, the analytical methods assume unrelatedness among population founders, yet instances where this assumption was found to be erroneous and leading to an overly positive assessment of genetic variation have been documented [31,32]. Pedigree-based genetic diversity parameters also fail to account for variation within the initial founder population. The “percentage genetic diversity loss”, which is a pedigree-based parameter commonly used to evaluate the success of conservation programs, may be misleading in cases where an originally diverse population experiences a higher percentage genetic diversity loss but retains higher absolute genetic variation than a population losing less diversity but from a less variable starting point [33]. Consequently, the inclusion of molecular data into captive management is recommended to enhance the accuracy of pedigree information [30,33,34,35,36]. In addition, molecular data might be used for the comparison of captive populations with their wild counterparts in order to assess how well the captive breeding programs represent total species diversity [23].

The purpose of this study was to examine the relationships between population history, management intensity, and genetic diversity in ungulate captive breeding programs in order to guide future genetic management strategies. We focused on the genetic diversity of six spiral-horned antelope taxa bred in the European Association of Zoos and Aquaria (EAZA) as follows: the mountain bongo (*Tragelaphus eurycerus*), nyala (*Tragelaphus angasii*), sitatunga (*Tragelaphus spekii*), lesser kudu (*Tragelaphus imberbis*), greater kudu (*Tragelaphus strepsiceros*), and common eland (*Tragelaphus oryx*). The captive populations of the selected taxa vary in terms of the available demographic and pedigree data, as well as the historic and current management. Moreover, the taxa vary considerably in their International Union for Conservation of Nature (IUCN) Red List statuses, wild population size, trends, and extant range size. Consequently, they represent a suitable and diverse set of cases for investigating the impact of these factors on the genetic diversity of captive populations.

The mountain bongo and lesser kudu, which are recognized as critically endangered and near-threatened, respectively, and exhibiting declining trends in their wild populations and restricted distributions [37,38], represent one end of the spectrum. Despite their EAZA populations being limited in size, their management is intensive and supported by a percentage of pedigree that is known to exceed 95% [39,40,41]. On the opposite end of the spectrum, the sitatunga and common eland are both identified as species of least concern with stable population trends and large wild ranges [42,43]. Although their EAZA populations are large in size, their management is based on incomplete pedigree records. Furthermore, these are the only two selected taxa without regularly published studbooks, and even herd management, where decisions are made for the whole herds rather than each individual separately, has only recently been abandoned by some institutions [41]. In the intermediate range between these two extremes are the nyala and greater kudu, which are both recognized as least concern with stable population trends [44,45]. Their EAZA populations are of a medium size, and although the known percentage of pedigree is only approximately 26%, studbooks are regularly published [41,46,47]. An overview of the parameters of the EAZA populations is provided in Table 1.

The specific aims of this study were to (a) compare the level of genetic diversity among the EAZA populations of the selected taxa and (b) evaluate the association of the obtained values with the aforementioned parameters differentiating the taxa in order to identify their impact on genetic diversity of the captive populations. We used a panel of 10 microsatellite loci to calculate basic descriptive parameters of genetic diversity for each of the populations. Sequences of the control region of the mitochondrial DNA of selected individuals were analyzed to determine geographic origin and genetic structuring within the populations.

## 2. Materials and Methods

Tissue, hair, and blood samples were collected from 20 zoos in 7 European countries (Austria, Czechia, Denmark, France, Germany, The Netherlands, the United Kingdom; Appendix A) and a captive breeding facility for common elands in Czechia between 2016 and 2019, with the exception of several older samples that had been obtained between 2000 and 2008.

Genomic DNA was extracted from tissue and hair samples using the DNeasy Blood and Tissue Kit (Qiagen, Venlo, The Netherlands), while the Genomic Blood/Cultured Cell DNA Mini Kit (Geneaid, New Taipei City, Taiwan) was applied for blood samples. The desired regions of DNA were amplified through polymerase chain reactions (PCRs) carried out in the T100 Thermal Cycler (BIO-RAD, Hercules, CA, USA).

To assess the descriptive parameters of genetic diversity, a panel of 10 microsatellite loci (BL42, BM4505, BRR, CSRM60, CSSM42, ETH10, ETH225, INRA011, INRA107, SPS113) was selected for genotyping (see Appendix A). PCR was performed in a reaction volume of 10 μL comprising 5 μL of Type-it Multiplex PCR Master Mix (Qiagen, Venlo, The Netherlands), 3 μL of RNase-free water, 1 μL of the primer mix (2 μM primer concentration), and 1 μL of the extracted DNA. After the 5 min initiation at 95 °C, 30 cycles of denaturation at 95 °C for 30 s, annealing at 60 °C for 90 s, and extension at 72 °C for 30 s were run, followed by a final extension of 30 min at 68 °C. Fragment analysis was carried out in a mixture of 8.5 μL of formamide, 0.5 μL of GeneScan 500 LIZ dye Size Standard (Thermo Fisher Scientific, Waltham, MA, USA), and 1 μL of PCR product in the Laboratory of Molecular Genetics (Faculty of Environmental Sciences, Czech University of Life Sciences Prague, Prague, Czech Republic). Genotypes were assembled from the obtained data following their editing and binning in Geneious 10.2.6 (https://www.geneious.com, accessed on 15 January 2023).

The control region of mitochondrial DNA was sequenced in selected individuals based on the ZIMS Species360 database [41] or studbooks [39,40,46,47] in order to cover the highest number of maternal lineages possible. The 25 μL PCR reaction mixture consisted of 12.5 μL of PPP Master Mix (Top-Bio, Vestec, Czech Republic), 8.5 μL of RNase-free water, 1 μL of MT4 primer [50], 1 μL of BT16168H primer [51], and 2 μL of the extracted DNA. The reaction conditions included an initiation at 95 °C for 5 min, followed by 29 cycles of denaturation at 95 °C for 1 min, annealing at 55 °C for 1 min, and extension at 72 °C for 1 min, and terminated through a 10 min final elongation at 72 °C. The PCR products were purified in case of successful amplification using the Gel/PCR DNA Fragments Extraction Kit (Geneaid, New Taipei City, Taiwan). Sequencing was performed in the service Seqlab laboratory at the Faculty of Science, Charles University. Obtained sequences were edited in Geneious 10.2.6 (https://www.geneious.com, accessed on 15 January 2023), and submitted to the National Center for Biotechnology Information (NCBI) Nucleotide database (see Appendix A for accession numbers).

Microsatellite data were used to calculate descriptive parameters of genetic diversity as follows: number of alleles (Na), number of effective alleles (Ne), observed heterozygosity (Ho), and expected heterozygosity (He) in GenAlEx 6.503 [52,53]. The inbreeding coefficient (Fis) and its 95% confidence interval (CI) were computed in Genetix 4.05 [54]. Additionally, Genepop 4.7.5 [55,56] was used to identify deviation from the Hardy–Weinberg equilibrium (HWE) by assessing heterozygote deficit and excess at a 0.05 level of significance.

Additional sequences were downloaded from NCBI nucleotide database, and their country of origin was determined based on the information provided in the original publications (Appendix A). For each studied taxon, the sequences were aligned using the ClustalW multiple alignment [57] and trimmed to reach an equal length in BioEdit Sequence Alignment Editor 7.2.5 [58]. The number of haplotypes (Nh), haplotype diversity (Hd), and nucleotide diversity (π) were determined using DnaSP 6.12.03 [59]. Neighbor-joining phylogenetic trees were constructed in MEGA11 [60] using the p-distance method as the number of base differences per site. The bootstrap values were reached by generating 500 replicates. FigTree 1.4.4. (http://tree.bio.ed.ac.uk/software/figtree/, accessed on 31 January 2023) was used for visualization and graphic editing of the phylogenetic trees. TCS [61] haplotype networks were created in PopART 1.7 [62].

## 3. Results

The amplification of the 10 microsatellite loci was successful across all the studied taxa except for the CSRM60 locus in the nyala and the CSSM42 locus in the lesser kudu. In the mountain bongo, a single allele was detected in 4 out of the 10 loci. Consequently, the mountain bongo population showed the lowest average number of alleles (2.10) and effective alleles (1.48) per locus in contrast with the highest values of 7.70 and 4.80 that were detected in the common eland (Table 2). The lowest number of effective alleles relative to the total number of alleles was observed in the nyala, with Ne only reaching 50% of Na, while this value ranged between 60% and 71% in the other taxa. The observed and expected heterozygosity were also lowest in the mountain bongo and the highest in the common eland. The nyala population displayed the smallest inbreeding coefficient (−0.119), whereas this parameter reached 0.260 in the mountain bongo population, which exceeded the value expected from reproduction of full siblings. Statistically significant deviation from HWE was only detected in two taxa—a heterozygote deficit in the mountain bongo and a heterozygote excess in the common eland (Table 2).

The number of haplotypes detected in the captive populations was lowest in the mountain bongo, where all the sampled individuals shared a single haplotype, and was highest in the common eland with 10 identified haplotypes. Haplotype diversity ranged from 0 in the mountain bongo to 0.8529 in the greater kudu, and nucleotide diversity varied from 0 in the mountain bongo to 0.183 in the greater kudu (Table 2). While shallow genetic differentiation was observed in the mountain bongo and lesser kudu, genetically distinct clades corresponding to geographic distribution patterns were identified in the nyala, sitatunga, greater kudu, and common eland (see Figure 1, and Appendix A). In all four species, the EAZA population was found to contain individuals from each of the naturally occurring extant clades; however, their distribution was substantially skewed towards one of them in the sitatunga, greater kudu, and common eland.

## 4. Discussion

This study provides an insight into the genetic state of the zoo populations of six spiral-horned antelope taxa where information is scarce as few genetic studies have been performed [63,64,65,66] and large proportions of pedigree are missing.

Genetic diversity of the captive populations reflects the impact of both historic and current management, although other factors, such as parameters of the wild populations, play a role. Despite being the most intensely managed and threatened taxa, the mountain bongo and lesser kudu showed different values of genetic diversity. The low genetic diversity of the mountain bongo population is in accordance with the findings of O’Donoghue et al. [63] and Combe et al. [67] and could be linked to the reduced variability of the subspecies in the wild, which was observed by Faria et al. [68] and is a common occurrence in many critically endangered taxa [69,70,71,72,73,74,75,76]. The founder population, despite being relatively large, may therefore have been of limited genetic diversity to start with. A similar situation was observed in the Grevy’s zebra (*Equus grevyi*) captive population [33], where the unexpectedly low variation in the founder base was most likely a consequence of the low genetic diversity present in the wild as the founders were sourced from a variety of locations. This cannot be said for the mountain bongo population as all of its founders were captured in the Aberdare Mountains, and no other areas of its distribution are thus represented in the captive population [77]. However, since information on the genetic composition of the wild population at the time of the capture of the founders is not available, it is impossible to predict the impact that the inclusion of founder animals from other localities would have had on the genetic state of the captive population. Nevertheless, it is safe to assume that an absolute absence of management would have resulted in even lower genetic diversity.

On the other hand, the lesser kudu population showed higher genetic diversity than the mountain bongo even though the intensity of management was comparable, which implies that other factors must have contributed to the degree of genetic diversity. The captive stock could have benefited from the larger wild population and its distribution area, plausibly resulting in higher variation in the captive founder base. We found a negative inbreeding coefficient in the lesser kudu population, which is in contrast with the value of 0.0941 estimated based on pedigree data [40]. This discrepancy could be linked to the fact that 35 out of the 38 analyzed samples had been collected in the Dvůr Králové Safari Park, and our result might thus only be representative of the animals present in this institution. However, this zoo holds more than half of the EAZA population, and a further 20% of the individuals either originate from this institution or are first- or second-generation offspring of such animals [40,41], which implies that this institution largely contributes to the genetic composition of the entire EAZA population.

Contrasting results were also found in the group of taxa in the middle of the spectrum. The genetic diversity parameters of the nyala population most closely resembled those of the mountain bongo, which could be a reflection of the low number of founders. The base of 13 founder animals is smaller than the minimum numbers generally recommended for retention of genetic diversity [23,78,79]. However, the low level of inbreeding implies that even though the population is of a low variability, reproduction of related individuals has been successfully prevented through the exchange of individuals among the zoos. In 2019, 30 animals, representing approximately 9% of the whole population, were reciprocally translocated among the EAZA zoos [47].

On the contrary, the greater kudu population showed relatively high genetic diversity, which could be associated with a larger wild distribution range and consequent variability compared with nyala. Similarly, the sitatunga and common eland populations displayed a high degree of genetic diversity despite the low intensity of management, which is likely to be linked to their large wild distribution and population size. Importantly, the presence of multiple genetically distinct lineages in all three of these zoo populations suggests that the observed genetic diversity parameters may have been ameliorated through the inclusion and mixing of genetically diverse founders. However, maintenance of captive population structures representative of their wild counterparts is one of the goals of captive management [80], and the interbreeding of distinct lineages may result in the genetic composition of the captive stocks that is different from that of the wild populations. For example, a relatively high percentage of admixed individuals has been observed in several captive populations, such as those of the common chimpanzee (*Pan troglodytes*) [81] and African dwarf crocodile (*Osteolaemus* spp.) [82]. Furthermore, the release of orangutan (*Pongo* spp.) individuals from distinct genetic lineages into one geographic area has caused their interbreeding and the likely dispersal of the admixed animals into other localities [83].

Although the interbreeding of distinct lineages has successfully been used to enhance genetic diversity in numerous cases [84,85,86,87,88,89,90], it is a highly debated topic due to its potential to lead to outbreeding depression [91,92,93,94,95]. In very small inbred populations, the precautionary principle is likely to support genetic mixing as the risks of inbreeding and genetic drift outweigh those of outbreeding depression and the disruption of locally adapted gene complexes [92,95]. However, in the case of the sitatunga, greater kudu, and common eland, wild populations are relatively large and are likely to possess a sufficient degree of genetic variation such that urgent genetic rescue is not required. Furthermore, the species occupy large and diverse habitats, thereby raising the likelihood of local adaptation. Consequently, the precautionary principle supports maintaining the integrity of the in situ population genetic structure within captive breeding management, which in turn requires genetic diversity to be sustained through other actions such as the exchange of animals from the same genetic lineage or the introduction of additional genetically appropriate founders. The admixture of distinct lineages and the associated outbreeding risk would thus be prevented at least until conditions change.

This study sheds light on the complexity of genetic management in zoo populations and provides evidence of the types of bias that can arise when applying well-documented assumptions to pedigree management in the absence of empirical data. However, it is worth acknowledging that our estimates of molecular genetic diversity in the EAZA populations of the spiral-horned antelope taxa are limited due to the application of small numbers of microsatellite and mitochondrial markers that only represent a fraction of the genome. Furthermore, we only used a limited number of individuals. Consequently, the inclusion of genome-wide markers, such as SNPs, and a larger proportion of the EAZA stocks would be beneficial to allow for a more in-depth analysis of the genetic state of these populations.

## 5. Conclusions

In this study, we compared genetic diversity among six spiral-horned antelope taxa within their EAZA populations. We evaluated the parameters of genetic diversity in the context of a variety of demographic and management parameters, revealing a connection with both present and historical management factors such as the captive population size and the number of founders. However, the results also reflect the variation present in wild populations, and in the critically endangered mountain bongo with an extremely low genetic diversity in the wild, this parameter appears to have a more pronounced impact than the management factors. We identified the presence of genetically distinct lineages in the captive populations of the sitatunga, greater kudu, and common eland. The relatively high levels of genetic diversity within these populations are most likely related to admixture among the originally differentiated lineages. Given their extensive natural distributions, the existence of local adaptations cannot be rejected, caution should be taken in interpreting the elevated diversity statistics, and the precautionary principle of maintaining natural population structure should be considered. The findings of this study support the importance of the inclusion of molecular data into captive population management. In addition to their usefulness in offering insight into the genetic condition of the populations where pedigree data are scarce or inaccurate, the results enable the identification of genetically complex situations, such as admixtures of distinct lineages, which are not always apparent from pedigree data.

## Figures and Tables

**Figure 1 biology-13-00104-f001:**
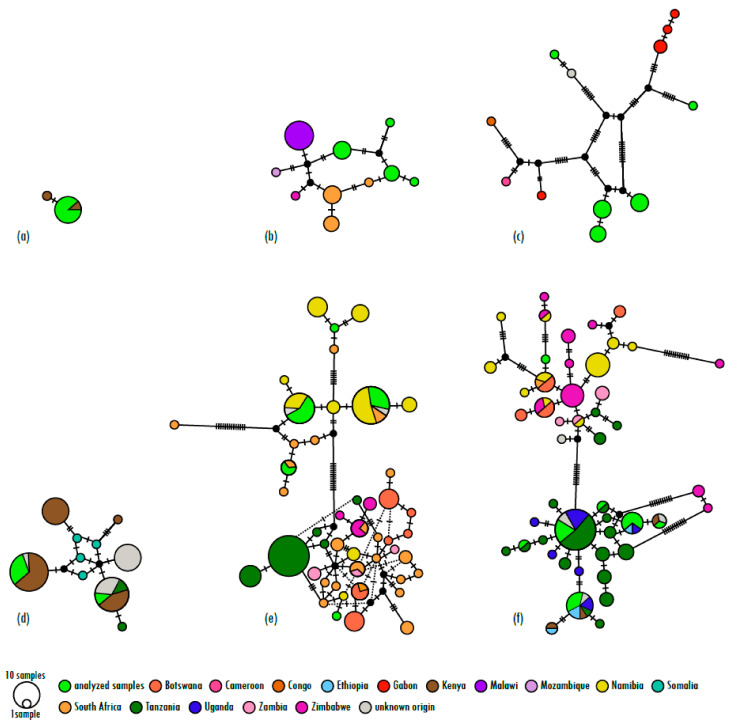
TCS haplotype networks for (**a**) the mountain bongo, (**b**) the nyala, (**c**) the sitatunga, (**d**) the lesser kudu, (**e**) the greater kudu, and (**f**) the common eland, including samples from this study and sequences downloaded from the National Center for Biotechnology Information (NCBI) nucleotide database. Geographic origin of the sequences is depicted through the different colors. The size of the circles is proportional to the number of sequences of the given haplotypes, mutation steps are indicated through hatch marks, and the length of lines is proportional to the number of mutation steps. Cases where proportionality could not be maintained are indicated through dotted lines.

**Table 1 biology-13-00104-t001:** Parameters of the European Association of Zoos and Aquaria (EAZA) populations of the studied taxa [37,38,39,40,41,42,43,44,45,46,47,48,49].

	Mountain Bongo	Nyala	Sitatunga	Lesser Kudu	Greater Kudu	Common Eland
number of captive animals ^1^	174	337	459	82	226	626
% of pedigree known ^2^	97%	26%	~45%	97.6%	26.4%	unknown
number of founders ^3^	33	13	25	24	24–25	unknown
IUCN status ^4^	CR	LC	LC	NT	LC	LC
studbook	yes	yes	no	yes	yes	no

^1^ is current as of 31 December 2022; ^2^ is current as of 2011 in the greater kudu, 2018 in the nyala, 2021 in the mountain bongo and lesser kudu, 2023 in the sitatunga; ^3^ is current as of 2003 in the sitatunga, 2011 in the greater kudu, 2018 in the nyala, 2021 in the mountain bongo and lesser kudu; ^4^ CR—critically endangered, NT—near-threatened, LC—least concern.

**Table 2 biology-13-00104-t002:** Values of basic descriptive parameters in the studied taxa.

	n_1_	Na	Ne	Ho	He	Fis (95% CI)	n_2_	Nh	Hd	π
Mountain bongo	10	2.10	1.48	0.164	0.208 *	0.260 (0.059 to 0.441)	8	1	0.0000	0.0000
Nyala	63	3.00	1.50	0.363	0.322	−0.119 (−0.189 to −0.063)	9	4	0.7500	0.0087
Sitatunga	24	4.80	2.89	0.612	0.618	0.034 (−0.111 to 0.118)	13	5	0.8077	0.0153
Lesser kudu	38	4.40	2.99	0.603	0.580	−0.026 (−0.120 to 0.046)	8	4	0.6429	0.0037
Greater kudu	17	5.10	3.28	0.676	0.635	−0.033 (−0.176 to 0.027)	17	9	0.8529	0.0183
Common eland	27	7.70	4.80	0.842	0.786 *	−0.051 (−0.127 to −0.018)	18	10	0.8301	0.0160

* indicates significant deviation from HWE at a 0.05 level of significance, n_1_—number of samples in microsatellite DNA analysis, Na—number of alleles, Ne—effective number of alleles, Ho—observed heterozygosity, He—expected heterozygosity, Fis—inbreeding coefficient, CI—confidence interval, n_2_—number of samples in mitochondrial DNA analysis, Nh—number of mitochondrial DNA haplotypes in the analyzed samples, Hd—haplotype diversity of the analyzed samples, π—nucleotide diversity.

## Data Availability

The mitochondrial DNA sequences were submitted to GenBank at the NCBI repository.

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
