# Peer review of "Evaluation of the Impact of Population Management on the Genetic Parameters of Selected Spiral-Horned Antelopes"

_biology, 2024, doi:10.3390/biology13020104_

Round 1
Reviewer 1 Report
Comments and Suggestions for Authors
This MS provide informative data on genetic diversity in captive populations of the spiral-horned antelope, with reference to their histories, and can be published after the following is attended to:
Methodology:
1. Where PCR reaction mixtures are reported, provide final concentrations of primers rather than volumes only.
2. For the analysis of mtDNA data, why not include nucleotide diversity to supplement haplotype diversity? The latter can be a bit conservative while nucleotide diversity covers all positions in the dataset.
Discussion:
1. The introductory part should place more emphasis on the fact that species have unique levels of genetic diversity. I understand why the authors are comparing values for species, but add a statement in the first paragraph to acknowledge that levels of diversity will also vary depending on evolutionary history.
2. Comparison of the study values with published values can be improved. At present, few absolute values are quoted in the discussion, rather there is reference to studies only. Ideally, it would have been valuable to see a table with values from the current study and previously published values for the species side-by-side, but perhaps there is not enough data of consistent format to do that. Nevertheless, it will improve the MS a lot if the statements on high or low diversity in natural populations are motivated with comparison of specific numbers.
Author Response
First of all, we would like to thank the reviewer for the time and effort invested in improving this manuscript. We found the comments valuable, and we have tried to adjust the text based on them. Are responses are highlighted.
- Where PCR reaction mixtures are reported, provide final concentrations of primers rather than volumes only.
Final concentration of primers has been included in the Methodology section.
- For the analysis of mtDNA data, why not include nucleotide diversity to supplement haplotype diversity? The latter can be a bit conservative while nucleotide diversity covers all positions in the dataset.
Nucleotide diversity has been added to Table 1 and Results section.
Discussion:
- The introductory part should place more emphasis on the fact that species have unique levels of genetic diversity. I understand why the authors are comparing values for species, but add a statement in the first paragraph to acknowledge that levels of diversity will also vary depending on evolutionary history.
The main aim of the comparison was to evaluate the impact of different factors on the current levels of genetic diversity in zoo populations. In this paragraph, we evaluate species with similar parameters, and we believe that our selected parameters serve as a proxy for species history.
- Comparison of the study values with published values can be improved. At present, few absolute values are quoted in the discussion, rather there is reference to studies only. Ideally, it would have been valuable to see a table with values from the current study and previously published values for the species side-by-side, but perhaps there is not enough data of consistent format to do that. Nevertheless, it will improve the MS a lot if the statements on high or low diversity in natural populations are motivated with comparison of specific numbers.
Studies on genetic diversity of zoo populations of spiral-horned antelope taxa are scarce and the number of studied parameters is limited. Research using microsatellite data has only been performed in mountain bongo, and studies using mitochondrial data have only been done in mountain bongo, lesser kudu, greater kudu, and common eland. No genetic studies of captive populations have been published for the populations of nyala and sitatunga. Furthermore, direct quantitative comparison with both types of marker is problematic unless data from exactly the same panel of microsatellites or mtDNA sequence region are available for comparison. The available data is thus insufficient to create such a comparison of the study values.
Reviewer 2 Report
Comments and Suggestions for Authors
Dear Authors,
the manuscript is of scientific interest and deserves acceptance for publication in the journal. I recommend improving it. I have provided recommendations in the text. Please make additions.

Author Response
First of all, we would like to thank the reviewer for the time and effort invested in improving this manuscript. We found the comments valuable, and we have tried to adjust the text based on them. Our replies are highlighted:
The summary should briefly mention the research methods used.
Specification of the used molecular methods has been added to the simple summary.
In their natural habitats, conservation and ongoing population monitoring strategies have been developed for various rare species to monitor population changes and make decisions (Morueta-Holme et al., 2010; Andreychev et al., 2019). But this is not equally possible for all species.
A phrase emphasizing the loss of biodiversity and genetic diversity has been added. Conservation of species in relation to the loss of biodiversity is further elaborated in the following paragraph. The insufficiency of in-situ conservation, especially in endangered species, was highlighted.
Specially protected natural areas and zoos play a significant role in this.
The role of zoos in conservation is explained in the following paragraph.
it is necessary to provide data on the number of individuals of specific animal species in different zoos. The names of the zoos must be indicated. It's better to do this in a separate table. This is important from the point of view of the distribution of the gene pool among different zoos. It is necessary to indicate the zoos in which the richest gene pool is concentrated. The authors do not provide information about the distance of zoos from each other. This is also important in terms of cooperation on the conservation of rare animal species.
The number of samples of each included taxon is provided in Table S1 in Supplementary Materials, which also contains information on the institutions of sample collection and their location. Inter-zoo comparisons of genetic parameters are beyond the aims of the research, which focused on the EAZA populations as whole. Each EAZA species program manages individuals across multiple zoo collections as a single genetic population. The distance among institutions is not particularly relevant, since different means of transport of animals among zoos are used, and distance measured for only one of them would not be relevant for the others.
Which zoos and countries?
The list of countries has been added to the text, and the list of zoos is provided in Table S1 in Supplementary Materials.
These results could be presented in relation to specific zoos. The authors can find an explanation for these data by analyzing the known relationships of animals from different zoos. In addition, it is advisable to indicate other characteristics of animals in the sample size (sex, age) for different species.
As this research was focused on EAZA populations as whole, rather than stocks present in specific institutions, this is out of the scope of this study. Each EAZA species program manages individuals across multiple zoo collections as a single genetic population. More zoo-specific details have been summarized in a report aimed at zoos and studbook keepers, which is currently under revision.
Based on the above conclusion, it is unclear what specific recommendations the authors offer for the conservation of these animal species based on their own results. It is clear that animals in captivity are different from natural populations. But what rational suggestions can be made for the practical application of the results?
This study was predominantly aimed at description and evaluation of genetic diversity present in the EAZA populations. As specific recommendations for management might be short-lived, they were not included in the conclusions of the manuscript. Recommendations for zoo population management have been proposed in a report aimed at zoos and studbook keepers, which is currently under revision.
Reviewer 3 Report
Comments and Suggestions for Authors
Author Response
First of all, we would like to thank the reviewer for the time and effort invested in improving this manuscript. We found the comments valuable, and we have tried to adjust the text based on them. Our replies are highlighted:
Conciseness: Condense sentences to be brief while retaining crucial information. For example, the research is specifically examining the genetic diversity of six species of spiral-horned antelope. The introduction adequately presents the necessary background information for readers, however it might be enhanced to have a more significant impact on the study.
Here are some suggestions to consider: Transition phrases should be included to facilitate a smoother progression of the reader's understanding across the many facets of the argument. Phrases such as "In this context" or "Moreover" may be used to establish connections between concepts.
We have added the suggested connectors throughout the text. We have also revised numerous sentences so that the overall manuscript is more concise.
Define Acronyms: Provide the definitions of acronyms such as EAZA and IUCN when they are first mentioned to ensure that readers are acquainted with these terms.
Upon revision, no cases of undefined acronyms have been found.
Connection to Purpose: Articulate the study's purpose and its contribution to tackling the identified difficulties in a clear manner. This will enhance readers' comprehension of the research's relevance. Capture the reader's attention by inserting a compelling remark or question that highlights the importance of the situation. This might enhance the allure of the introduction
We have now added a clear statement of purpose: “The purpose of this study was to examine the relationships between population history, management intensity and genetic diversity in ungulate captive breeding programs, in order to guide future genetic management strategies.”